# High-Frequency Basal Cell Carcinoma: Demographic, Clinical, and Histopathological Features in a Belgian Cohort

**DOI:** 10.3390/jcm14134678

**Published:** 2025-07-02

**Authors:** Katharina Charlotte Wunderlich, Carmen Orte Cano, Mariano Suppa, Olivier Gaide, J. M. White, Hassane Njimi, Véronique Del Marmol

**Affiliations:** 1Department of Dermatology, Hôpital Erasme, Université Libre de Bruxelles, 1070 Brussels, Belgium; katharina.wunderlich@hubruxelles.be (K.C.W.);; 2Department of Dermatology and Venereology, CHUV, University of Lausanne, 1011 Lausanne, Switzerland

**Keywords:** high-frequency basal cell carcinoma (HF-BCC), basal cell carcinoma (BCC), non-melanoma skin cancer (NMSC), squamous cell carcinoma (SCC), EUSCAP platform

## Abstract

**Background**: Basal cell carcinoma (BCC) is the most common skin cancer worldwide, with a multifactorial aetiology involving environmental and intrinsic factors. A small subset of patients develops high-frequency BCC (HF-BCC), defined as ≥9 BCCs within 3 years. **Objective**: To analyse demographic, clinical, and histopathological features of non-syndromic HF-BCC in a Belgian cohort, compared with low-burden BCC patients and healthy controls. **Methods**: A retrospective cohort study was conducted at Erasme Hospital (Brussels) using data from the EUSCAP platform. Clinical, behavioural, and histopathological data were collected and statistically analysed. **Results**: Of 783 patients, 16 with HF-BCC were identified. For comparison, 32 patients with 1–2 BCCs and 117 patients without BCC were selected. HF-BCC patients showed distinct characteristics, including a higher proportion of superficial BCCs (68.3% vs. 50%, *p* = 0.01) and fewer nodular subtypes (43.2% vs. 63.5%, *p* = 0.01). Their tumours were less frequently located on the nose and ears compared with patients having 1–2 BCCs. HF-BCC was associated with a personal history of squamous cell carcinoma (SCC) and actinic keratosis (AK). **Conclusions**: HF-BCC patients display distinct anatomical, histopathological and clinical characteristics, with a predominance of superficial BCC and an association with a personal history of SCC and AK. They show a lower frequency of tumours on the nose and ears, with a stronger tendency for localisation on the trunk and extremities. Identifying risk factors and genetic markers may contribute to improved early detection strategies, preventive measures, and the development of targeted therapies.

## 1. Introduction

Basal cell carcinoma (BCC) is the most common skin cancer worldwide, accounting for over 85% of all NMSCs in Europe [1,2,3]. Its development results from a multifactorial interplay of environmental and individual predispositions.

Extrinsic risk factors for BCC include ultraviolet (UV) radiation from both natural and artificial (sunbeds, medical radiation) sources, lifestyle (consumption of spirits) and medical factors, such as immunosuppression. Intrinsic risk factors for BCC comprise Fitzpatrick phototypes I-III, light hair and eye colour, as well as childhood freckles. A personal history of non-melanoma skin cancer (NMSC), male sex and increasing age are also associated with BCC. Lastly, BCC pathogenesis has been linked to several genetic alterations, including mutations in the Hedgehog pathway (Patched 1 (PTCH1), Smoothened (SMO]), the tumour suppressor Tumour Protein p53 (TP53), proto-oncogenes of the Rat Sarcoma (RAS) family, and germline variants in the Melanocortin 1 Receptor (MC1R) gene [4,5].

A small subset of patients develops multiple BCCs within a few years, termed high-frequency BCC (HF-BCC). The most commonly used definition of HF-BCC in the recent literature is the diagnosis of 9 or more BCCs within a 3-year period [6,7]. While some cases are linked to known genetic syndromes like basal cell nevus syndrome (Gorlin-Goltz—PTCH1 mutation) or xeroderma pigmentosum (defects in nucleotide excision repair genes), non-syndromic HF-BCC remains poorly characterized [8,9]. A retrospective study comprising 4943 BCCs and affecting a total of 2407 patients in America found that among patients with non-syndromic BCCs monitored for 5 to 10 years, 3% of patients developed 9 or more BCCs [10]. A retrospective cohort study of the Danish health national registries found that the prevalence of patients with non-syndromic HF-BCC during any 3-year period from 1999 to 2013 was 49.39 per 100,000 [6]. Furthermore, HF-BCC was found to be associated with male sex, a personal history of squamous cell carcinoma (SCC), and melanoma. Patients with HF-BCC were also noted to have a 4-fold higher risk for other malignancies, such as lymphoma, leukaemia, breast and colon cancer. However, further research needs to be done for stratifying patients at highest risk for developing subsequent BCCs [6,7,10,11].

The present study aimed to investigate HF-BCC patient characteristics by analysing demographic, clinical, and histopathological and behavioural data for the first time in a Belgian patient cohort. Through comparison with individuals presenting with low BCC burden and healthy controls, the goal was to identify risk factors and anatomical patterns that may inform future risk stratification and clinical management in non-syndromic HF-BCC.

## 2. Methods

### 2.1. Data Collection

A retrospective cohort study was conducted at the Dermatology Department of Erasme Hospital in Brussels, Belgium. We used data from the EUropean Skin CAncer risk factor Platform (EUSCAP) that encompasses demographic, constitutional, behavioural, clinical, and histopathological information. The database was constructed using questionnaires completed by patients undergoing skin cancer screenings at the dermatology department [12,13]. At the time of analysis, the database included records of 783 patients collected from 2021 to 2024. For each patient, data were collected on age, sex, BMI, ethnicity, smoking status (current and former), cumulative sun exposure (occupational, recreational, intentional), history of sunburns, sunbed use, and residence in high UV-exposure countries. Constitutional variables included Fitzpatrick phototype, natural hair and eye colour, childhood freckles, and the presence of AKs, solar lentigines, congenital and atypical naevi, as well as total naevi count. Clinical history encompassed history of personal and familial skin cancer, including melanoma and non-melanoma types. In addition, histological subtypes and anatomical localisations of all BCCs were systematically recorded. Patients with genetic syndromes or immunosuppression (e.g., organ transplant recipients) were excluded.

### 2.2. Statistical Analysis

Statistical analysis was conducted to compare the clinical, histological, and anatomical characteristics of patients with HF-BCC, those with 1–2 BCCs, and control subjects without BCC, all within the same age range (43–91 years). Continuous variables are presented as mean ± standard deviation (SD), while categorical variables are reported as counts and percentages (*n* (%)). Group comparisons were performed using the t-test for continuous variables and either Pearson’s Chi-square test with continuity correction or Fisher’s exact test for categorical variables, as appropriate. All tests were two-sided, and a *p*-value of <0.05 was considered statistically significant. Statistical analyses were carried out using SPSS for Windows, version 28.0 (Chicago, IL, USA).

## 3. Results

### 3.1. Demographics, Clinical Characteristics and Sun Exposure in HF-BCC Patients

Out of 783 patients, 16 met the inclusion criteria for HF-BCC (2%) and 32 for 1–2 BCCs (4.1%). All of them were of white/Caucasian ethnicity. With an increasing number of BCCs, the proportion of males rose progressively from 44.4% in patients without BCC to 46.9% in patients with 1–2 BCCs and reaching 50% in HF-BCC patients. Patients with 1–2 BCCs were significantly older than healthy controls (70.19 years ± 11.7 vs. 62.04 years ± 10.7, respectively, *p* = 0.001). However, the age difference between HF-BCC (67.63 years ± 14.6) and the other groups was not statistically significant. Regarding the age at first BCC diagnosis, patients with HF-BCC were on average younger (60 years ± 13.4) than patients with 1–2 BCCs (65.84 years ± 14) (Table 1).

When analysing behavioural risk factors, we observed that HF-BCC patients tend to report more frequent intentional sun exposure, with 75.0% of the HF-BCC group and 68.8% of the 1–2 BCC group stating they had sunbathed more than 25 times in their lifetime, compared with 59.8% in the control group without skin cancer (*p* = 0.28). Furthermore, a higher proportion of HF-BCC patients reported having spent one year or more in a country with high sun exposure, with 50.0% of individuals in the HF-BCC group and 37.5% in the 1–2 BCC group, compared with 28.2% in control group (*p* = 0.17) (Table 1).

Concerning the clinical findings, there was a trend towards a higher prevalence of solar lentigines with increasing BCC burden: 69.6% in controls, 71.9% in the 1–2 BCC group, and 81.3% in HF-BCC patients (*p* = 0.62). Our findings reveal a gradient in the prevalence of AKs, which increases progressively with rising BCC burden, ranging from 22.4% in patients with no BCCs to 37.5% in those with 1–2 BCCs and reaching 50.0% in patients with HF-BCC, which was significantly higher than in individuals without BCC (*p* = 0.03). A significantly higher personal history of SCC was observed in HF-BCC patients (25%) and patients with 1–2 BCCs (31.3%), compared with the control group (6.8%, *p* = 0.001) (Table 1).

### 3.2. Sun Exposure and Family History of NMSC

In the HF-BCC patient group, cumulative total sun exposure ranged from 164 to 3964 weeks, with annual exposure varying between 3 and 50 weeks. Four participants reported a positive family history of NMSC, while ten had no such history. Individual-level data on sun exposure and family history of NMSC in the HF-BCC cohort revealed notable variation across participants. For instance, several older individuals (aged 79–91 years) reported very high cumulative (2856–3964 weeks) and annual (34–50 weeks) sun exposure, while indicating no family history of skin cancer. In contrast, other individuals had far lower levels of cumulative (219–328 weeks) and annual (3 weeks) sun exposure, but a positive family history. These findings highlight the heterogeneity of sun exposure habits and genetic predisposition within the HF-BCC group (Table 2).

### 3.3. Histopathological and Anatomical Characteristics of HF-BCC

Regarding the different histological subtypes of BCC, significant differences were observed between HF-BCC patients and those with 1–2 BCC. HF-BCC patients were more likely to develop superficial BCCs (343 of 502 (68.3%) of all BCCs in this group) than patients with 1–2 BCCs (26 of 52 (50%), *p* = 0.01). Conversely, HF-BCC patients showed a lower rate of nodular BCC (43.2%) compared with patients with 1–2 BCCs (63.5%, *p* = 0.01). The infiltrative BCC subtype was more frequent in patients with 1–2 BCCs (21.2%) than in HF-BCC patients (12.5%). However, this trend did not reach statistical significance (*p* = 0.08) (Table 3). Only 2.2% of all BCCs were classified as histological subtypes other than nodular, superficial or infiltrative (i.e., micronodular, morpheiform, basosquamous, adenoid etc.) with similar proportions across both groups (3.8% in patients with 1–2 BCCs vs. 2.0% in patients with HF-BCC).

Furthermore, the two groups exhibited distinct patterns in the anatomical distribution of the BCCs, with specific anatomical regions showing varying frequencies: BCCs in HF-BCC patients were less frequently located on the nose (2%) and ears (2.4%) compared with patients with 1–2 BCCs, where the respective frequencies were 7.7% (*p* = 0.01) and 7.7% (*p* = 0.03). On the other hand, HF-BCC patients showed a tendency to develop BCC more frequently on the trunk (37.3%, *p* = 0.1) and the extremities (23.1%, *p* = 0.2) than patients with 1–2 BCCs (26.9%, and 15.4% respectively), although these observations did not reach statistical significance (Table 4).

## 4. Discussion

In our dermatological outpatient cohort, 2% of patients developed HF-BCC. In contrast, studies based on national registries and insurance claims covering the general population have reported prevalences of approximately 0.05% in the United States and Denmark [6,7]. The markedly higher proportion of HF-BCC patients observed in our dermatological outpatient cohort compared with the prevalence reported in population-based studies suggests a selection bias inherent to specialized clinical settings.

The findings of this study should be interpreted with caution due to the limited sample size of HF-BCC patients (*n* = 16). Furthermore, interpretation of the findings should consider the complex interplay of genetic and environmental risk factors, as it remains unclear whether HF-BCC represents a distinct clinical entity or lies along a continuous risk spectrum for BCC. Nonetheless, this study provides important preliminary insights into this rare clinical phenotype in a Belgian cohort.

Upon closer examination of this patient group, we showed that HF-BCC individuals display characteristic clinical, histological, and anatomical patterns compared with patients with low BCC burden or no history of skin cancer. In line with existing literature, we observed a higher personal history of SCC in patients with 1–2 BCCs and HF-BCC [6,7]. A former NMSC is a well-known risk factor for BCC. The 3-year cumulative risk of developing an NMSC of any type after a first NMSC ranges from 35% to 60%. The mean 3-year cumulative risk of developing a BCC for patients with a history of an SCC is about 43% [14,15]. Concerning patients with HF-BCC, a national retrospective cohort study from the United States found that having 9 or more BCCs versus only 1 BCC during the 3 years was associated with a history of SCC, male sex, and melanoma [7].

We described for the first time a gradient in the prevalence of AKs, with HF-BCC patients having significantly more AKs than patients without BCC.

AK is a precancerous skin lesion caused by chronic UV exposure, typically affecting sun-exposed areas in older adults. It can present with multiple lesions and carries a variable but low annual risk (0–0.53% per lesion) of progressing to SCC, reaching 2.88% at 5 years [16,17,18]. The increased occurrence of HF-BCC in patients with AK and a history of SCC may reflect a cumulative effect of UV exposure and shared pathomechanisms underlying the development of these skin lesions. Recently, genome-wide association (GWA) meta-analyses have identified over 120 BCC-associated loci, enhancing our understanding of the genetic basis of BCC susceptibility. Moreover, a high degree of association of these BCC-associated loci with SCC risk was observed. Some of these loci were related to pigmentation like SLC45A2, TYR, OCA2 or MC1R. This genetic overlap could explain the association between HF-BCC and personal history of SCC in our study cohort [19]. Further genetic research in HF-BCC patients is needed to identify whether there are associated loci.

Analysis of sun exposure patterns and family history within our cohort of 16 HF-BCC patients suggests that this group is heterogeneous, comprising individuals whose high BCC burden is caused by different aetiologies, from excessive UV exposure to underlying genetic predisposition. Individual-level data on sun exposure and NMSC family history support this hypothesis: while some patients reported extensive lifetime sun exposure without a familial background of skin cancer, others developed numerous BCCs despite relatively limited sun exposure but had a positive family history. These findings suggest that both extrinsic (UV-related) and intrinsic (genetic) risk factors contribute to HF-BCC development, with the relative influence varying among individuals. Thus, the development of multiple BCCs in non-syndromic individuals likely results from cumulative risk factors rather than a singular underlying biological process.

We observed significant differences in histological subtypes and anatomical locations of BCC between HF-BCC patients and those with 1–2 BCC: HF-BCC patients more frequently developed superficial BCCs, while nodular BCCs were more common in patients with 1–2 BCCs. In terms of anatomical distribution, BCCs in patients with 1–2 BCCs were more often located on the nose and ears (*p* < 0.05), whereas HF-BCC patients showed a tendency for BCCs on the trunk (*p* = 0.1) and extremities (*p* = 0.2), although the latter observations did not reach statistical significance. These findings align with previous research indicating that the superficial BCC subtype is more commonly found on photoprotected areas like the trunk, whereas nodular and infiltrative BCC is located mainly on the face [2]. One hypothesis is that chronic cumulative sun exposure is more related to the nodular type of BCC (similarly to SCC) whereas intermittent sun-exposure is linked to superficial BCC [2,20].

These observations are consistent with research findings from Stanford, which stated that patients with 6 or more BCCs are more likely to have BCCs on the trunk compared with those with a single BCC (*p* = 0.011). Similarly, as the number of BCCs increases, they are more likely to be of the superficial subtype as compared with the subtypes in patients with a single BCC (*p* = 0.007) [10].

We observed a higher proportion of superficial BCCs and a reduced involvement of chronically UV-exposed facial regions in HF-BCC patients. This may be influenced by specific UV exposure patterns, such as intermittent sun exposure, or genetic peculiarities related to the tumour itself or patient-specific factors. The divergence in tumour distribution and subtype may reflect underlying molecular heterogeneity depending on tumour location: Site-specific mutational landscapes have been found to influence tumour growth patterns and histological subtypes in BCC. Several tumour suppressor genes and proto-oncogenes have long been known to be implicated in BCC pathogenesis, including key components of the Hedgehog pathway as PTCH1 and SMO, the TP53 tumour suppressor, and members of the RAS proto-oncogene family [4]. A recent study linked some of these mutations with the BCC clinic-pathological features [21]. They found a significant association between the superficial type of BCC and PTCH1 (OR = 5.537, 95% CI = 1.367–22.43) and Notch Receptor 1 (NOTCH1) (OR = 4.457, 95% CI = 1.304–15.24) mutations. In addition, PTCH1 mutations were found to be linked to intermittent sun exposure (*p* = 0.046), while NOTCH1 mutations were more frequent in BCCs located on the trunk compared with the head/neck and extremities (*p* = 0.001) [21].

Additionally, variations in the extracellular matrix composition can affect tumour behaviour by altering tissue mechanics, further impacting the tumour’s development: Bansaccal et al. proposed and tested the hypothesis that the different anatomical distribution of BCCs depending on their subtype could be explained by the influence of regional tissue environments on tumour initiation. They demonstrated that the oncogenic activation of SMO via a constitutively active mutant (SmoM2) leads to different tumour behaviour depending on the anatomical site. In mice, SmoM2 expression in the ear epidermis triggered clonal expansion, tumour initiation, and dermal invasion, whereas in the back-skin epidermis, the same oncogenic signal led only to lateral clonal expansion without invasive tumour formation. This regional disparity was linked to differences in extracellular matrix (ECM) composition and stiffness, particularly the density of collagen I. Notably, reducing collagen I levels or inducing structural changes via UV exposure or ageing made back-skin cells more permissive to tumour formation [22].

These findings support the hypothesis that the anatomical microenvironment, including ECM composition and mechanical properties, plays a pivotal role in modulating the tumorigenic potential of known oncogenic mutations of BCC, such as those affecting SMO. This could explain why certain body regions—particularly those with less permissive dermal architecture like the back—tend to develop non-invasive, superficial BCCs rather than deeply infiltrative nodular subtypes.

Our findings raise the question of whether histopathological patterns observed in non-syndromic HF-BCC differ from those seen in syndromic cases such as basal cell nevus syndrome or xeroderma pigmentosum. Indeed, studies in patients with basal cell nevus syndrome have shown a predominance of the nodular subtype, accounting for 63.1% to 78.7% of all BCCs [23,24,25]. In an Indian cohort of patients with xeroderma pigmentosum, the nodular BCC subtype was likewise reported as the most prevalent (41.46%) histopathological form among these syndromic cases [26]. This highlights a difference, with the superficial subtype being predominant in our non-syndromic HF-BCC patients. These observations may suggest distinct underlying pathophysiological mechanisms between syndromic and non-syndromic HF-BCC patients.

In conclusion, HF-BCC represents a clinical phenotype characterized by a high tumour burden. Still, the distinctiveness of the HF-BCC group should be interpreted with caution, and the concept of an HF-BCC threshold (≥9 BCCs within 3 years) warrants careful interpretation. It remains uncertain whether this threshold reflects a biologically meaningful demarcation or represents an arbitrary point along a continuous clinicopathologic spectrum. While the current study presents arguments for specific features in this patient subgroup, existing evidence does not robustly support the notion of unique clinical or molecular characteristics. Instead, available data suggests that non-syndromic HF-BCC is a combination of different patients characteristics: The occurrence of multiple BCCs in non-syndromic patients is linked to fair skin and family history as well as a variety of genetic characteristics (germline mutations in DNA repair genes and increased malignancy risk), rather than clearly indicating one single pathobiological process [11,27].

Investigating non-syndromic HF-BCC cases may help to better understand the interplay of various risk factors and pathological mechanisms, ultimately enabling more personalized treatment approaches and early identification of individuals affected by multiples BCCs.

## Figures and Tables

**Table 1 jcm-14-04678-t001:** Demographics and clinical characteristics. This table compares individuals without BCC (*n* = 117), with 1–2 BCCs (*n* = 32), and with HF-BCCs (*n* = 16). Variables include demographic factors (e.g., sex, age), indicators of sun exposure, and coexisting skin conditions or personal history of skin cancers. Superscript letters (a, b) denote statistically significant pairwise differences based on post hoc analyses. Abbreviations: *n*, number; %, percentage; n.a., not applicable; sd, standard deviation; BCC, basal cell carcinoma; HF-BCC, high-frequency basal cell carcinoma; SCC, squamous cell carcinoma.

	0 BCC (*n* = 117)	1–2 BCCs (*n* = 32)	HF-BCC (*n* = 16)	*p*-Value
Male, *n* (%)	52 (44.4)	15 (46.9)	8 (50)	0.901
Age (years), mean (sd)	62.04 ^a^ (10.7)	70.19 ^b^ (11.7)	67.63 ^a,b^ (14.6)	0.001
Age at first BCC (years), mean (sd)	n.a.	65.84 (14)	60 (13.4)	0.173
White/Caucasian, *n* (%)	112 (95.7)	32 (100)	16 (100)	0.909
≥1 year in country with higher sun exposure, *n* (%)	33 (28.2)	12 (37.5)	8 (50)	0.166
Sun tanning >25 times (lifetime), *n* (%)	70 (59.8)	22 (68.8)	12 (75)	0.277
Presence of solar lentigines, *n* (%)	80 (69.6)	23 (71.9)	13 (81.3)	0.624
Presence of actinic keratoses, *n* (%)	26 (22.4) ^a^	12 (37.5) ^a,b^	8 (50) ^b^	0.029
Personal history of melanoma, *n* (%)	20 (17.1) ^a^	11 (34.4) ^b^	4 (25) ^a,b^	0.098
Personal history of SCC, *n* (%)	8 (6.8) ^a^	10 (31.3) ^b^	4 (25) ^b^	0.001

**Table 2 jcm-14-04678-t002:** Cumulative and annual sun exposure, as well as family history of NMSC in HF-BCC patients. The table presents individual-level data on age, cumulative lifetime sun exposure (in weeks), average annual sun exposure (weeks per year), and the presence of a family history of NMSC. Abbreviations: n.d., no data; NMSC, non-melanoma skin cancer.

Age	Cumulative Lifetime Sun Exposure (Weeks)	Average Annual Sun Exposure (Weeks/Year)	Family History of NMSC
64	164	3	no
76	218	3	no
76	219	3	yes
47	294	6	no
43	328	8	yes
56	786	14	No
55	960	17	Yes
76	974	13	-
84	1084	13	no
62	1568	25	yes
85	2856	34	no
91	3675	40	no
79	3964	50	no
50	n.d.	n.d.	no
63	n.d.	n.d.	n.d.
75	n.d.	n.d.	no

**Table 3 jcm-14-04678-t003:** Histological subtype distribution of BCC in patients with 1–2 vs. HF-BCC. The table compares the distribution of histological BCC subtypes between patients with 1–2 BCCs (*n* = 52) and those with HF-BCCs (*n* = 502). BCC, basal cell carcinoma; HF-BCC, high-frequency basal cell carcinoma; *n*, number; %, percentage.

	BCCs in Patients with 1–2 BCCs (*n* = 52)	BCC in Patients with HF-BCC (*n* = 502)	*p*-Value
Superficial BCC, *n* (%)	26 (50)	343 (68.3)	0.008
Nodular BCC, *n* (%)	33 (63.5)	217 (43.2)	0.005
Infiltrative BCC, *n* (%)	11 (21.2)	63 (12.5)	0.083

**Table 4 jcm-14-04678-t004:** Anatomical distribution of BCC in patients with 1–2 vs. HF-BCC. The table shows the frequency of BCCs at various anatomical sites in patients with 1–2 BCCs (*n* = 52) compared with those with HF-BCCs (*n* = 502). BCC, basal cell carcinoma; HF-BCC, high-frequency basal cell carcinoma; *n*, number; %, percentage.

	BCCs in Patients with 1–2 BCCs (*n* = 52)	BCC in Patients with HF-BCC (*n* = 502)	*p*-Value
Nose, *n* (%)	4 (7.7)	10 (2)	0.013
Ears, *n* (%)	4 (7.7)	12 (2.4)	0.030
Face (any localisation), *n* (%)	20 (38.5)	134 (26.7)	0.071
Trunk, *n* (%)	18 (34.6)	220 (43.8)	0.202
Extremities, *n* (%)	8 (15.4)	116 (23.1)	0.203

## Data Availability

The data that support the findings of this study are available from the corresponding author upon reasonable request.

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
