# Peer review of "High-Frequency Basal Cell Carcinoma: Demographic, Clinical, and Histopathological Features in a Belgian Cohort"

_jcm, 2025, doi:10.3390/jcm14134678_

Round 1

Reviewer 1 Report

Comments and Suggestions for Authors

This paper dealing with characteristics of high-frequency basal cell carcinoma is presented well and of interest in particular for dermatologists; the table, the main data are clear and easy to read.

There are some points of criticism:

  1. High-frequency basal cell carcinoma is defined as nine or more BCC within three years. The distinction between these patients and syndromic patients, who also develop multiple BBCs, is missing but important and therefore should be added in some detail.
  2. Are there differences in histological and anatomical patterns between high-frequency BCC and syndromic patients? These data should be added.
  3. The conclusions drawn should be interpreted with more caution because only 16 patients have been included.

Author Response

Dear Reviewer,

Thank you very much for taking the time to review our manuscript. We are very grateful for your comments and remarks. Please find the detailed responses below and the corresponding revisions/corrections highlighted/in track changes in the re-submitted files. 

This manuscript has been re-reviewed for prose style and clarity by one of the authors who is a native English speaker.

Comment 1: High-frequency basal cell carcinoma is defined as nine or more BCC within three years. The distinction between these patients and syndromic patients, who also develop multiple BBCs, is missing but important and therefore should be added in some detail.
Answer: We address this in the lines 50-55: A small subset of patients develops multiple BCCs within a few years, termed high-frequency BCC (HF-BCC). The most commonly used definition of HF-BCC in the recent literature is the diagnosis of 9 or more BCCs within a three-year period6,7. While some cases are linked to known genetic syndromes like basal cell nevus syndrome (Gorlin-Goltz - PTCH1 mutation) or xeroderma pigmentosum (defects in nucleotide excision repair genes), non-syndromic HF-BCC remains poorly characterized.
Non-syndromic HF-BCC compromises all patientes with HF-BCC without known / described mutations like seen in Gorlin Goltz or Xeroderma pigmentosum.

Comment 2: Are there differences in histological and anatomical patterns between high-frequency BCC and syndromic patients? These data should be added.
Answer: Thank you very much for this excellent remark. We added a new section for this in the lines 272-281.

Comment 3: The conclusions drawn should be interpreted with more caution because only 16 patients have been included.
Answer: We added a sentence higlighting this limitation at the beginning of the discussion: While the findings of this study should be interpreted with caution due to the limited sample size of HF-BCC patients (n = 16), they nonetheless provide important preliminary insights into this rare clinical phenotype in the Belgian population.

Please find attached the corrected text. We are open and grateful for any further comments and suggestions!

Reviewer 2 Report

Comments and Suggestions for Authors

The manuscript described some demographic, clinical and histopathological features of high frequency BCC among a Belgian cohort.  The manuscript is well written however, it is not clear why the authors were looking at AK only while investigating BCC as it is not a precursor lesions for this skin cancer.  It would have been interesting had SK also be included in this study on the incidence of HF-BCC in this cohort of patients.

Author Response

Thank you for your valuable and insightful comments. You are absolutely right in noting that AK are not considered precursor lesions for BCC. AKs were included in our analysis because they are part of the EUSCAP questionnaire, which is designed to assess general skin cancer risk factors. As such, AKs were recorded in our database and could be evaluated in the context of High-Frequency BCC. Unfortunately, seborrheic keratoses (SK) were not captured by the EUSCAP questionnaire, and thus data on SK were not available for inclusion in our study - something we also regret a lot, as it would indeed have added another interesting dimension to our analysis.

Reviewer 3 Report

Comments and Suggestions for Authors

A well-written review and statistical analysis of the somewhat overlooked category of HF-BCC.

Very insightful and interesting to read.

I would like to propose some minor revisions and notes:

Lines 142–149: Very weak analysis regarding tumor subtypes. Have you merged them in any way? (Where are the basosquamous, micronodular, sclerotic, morphea-like, adenoid-like, follicular-like types...?)

Have you considered that many of the superficial spreading BCCs in the same anatomical region could be part of the same tumor?

Line 154: BCCs — please clarify the table. Also, is there no mention of the basosquamous type (again)?

Line 233: You should mention these points in line 45.

Author Response

Comment 1: 
Lines 142–149: Very weak analysis regarding tumor subtypes. Have you merged them in any way? (Where are the basosquamous, micronodular, sclerotic, morphea-like, adenoid-like, follicular-like types...?)
Thank you very much for your comment regarding the histopathological subtypes and the potential merging of categories. For the purpose of analysis, we recorded whether a specific subtype was present or not. However, we acknowledge that mixed histological patterns are common in BCC. In our analysis, the majority of BCC cases presented (at least partly - when mixed) subtypes such as nodular, superficial or infiltrative. Only a very small proportion (12 out of 554 cases (2 % in HF-BCC patients and 3.8% in BCC patients) were classified as “other” histological subtypes (e.g., basosquamous, adenoid-like, morpheaform). This small number did not allow for meaningful statistical subgroup analysis of these rarer subtypes. We added a section precising this.

Have you considered that many of the superficial spreading BCCs in the same anatomical region could be part of the same tumor?
Based on photographic documentation and histopathological analyses with margin control, we are confident that the lesions recorded as individual superficial spreading BCCs represent distinct tumours rather than manifestations of the same lesion. However, we acknowledge that a field cancerisation effect—similar to what is observed in actinic keratoses (AKs)—cannot be entirely ruled out.

Comment 2:
Line 154: BCCs — please clarify the table. Also, is there no mention of the basosquamous type (again)?
We have added a sentence above Table 3 to explain the handling of less common histological subtypes, including basosquamous BCC.

Comment 3:
Line 233: You should mention these points in line 45.
Thank you for the suggestion. We have now added the relevant points mentioned in line 233 to line 45, as recommended.

Thank you again for your constructive feedback - please find the revised manuscript attached. We remain open to any further suggestions or comments you may have.

Reviewer 4 Report

Comments and Suggestions for Authors

1 The main research question:  High Frequency BCC (HFBCC) is defined as greater than 9 cases over 3 consecutive years. This paper aims at identifying a range of associated risk factors and characteristics for HFBCC.  

2 The topic is relevant at the clinical coalface. Previous work of a similar and or related  nature is corrected cited in the Introduction and Discussion. The association of superficial BCC (more on trunk and limb sites) with HFBCC and more nodular and infiltrating BCC (head and neck sites) with non HFBCC is useful and consistent with earlier work. 

3 The content adds to previous published work. These Belgian cohort findings could be easily extended to other similar cohorts. 

4 No specific improvements in methodology are required. The paper is sound, well written and the findings clearly displayed in the Tables submitted.

5 The conclusions are consistent with the data provided. The discussion is concise, clear, reads well and is technically correct in detail.

6 The references are appropriate for the content.

7 The tables are clear and easy to follow. No Figures are part of the paper. 

Author Response

Thank you very much for your thorough and positive evaluation of our manuscript. We greatly appreciate your comments and are pleased that you find the study design, analysis, and presentation of results to be clear, relevant, and methodologically correct. We are especially grateful for your remarks on the clinical relevance and the consistency of our findings with previous work.

Reviewer 5 Report

Comments and Suggestions for Authors

The current study explores high-frequency basal cell carcinoma (HF-BCC), defined as the development of multiple BCCs—typically six or more within a 10-year period. HF-BCC can occur in both syndromic and non-syndromic forms, each associated with distinct etiologies and clinical profiles. According to your findings, HF-BCC patients exhibit specific features, including a higher proportion of superficial BCCs, fewer nodular subtypes, a lower frequency of tumors on the nose and ears, and associations with personal histories of squamous cell carcinoma (SCC) and actinic keratosis (AK).

To enhance the quality and clarity of the manuscript, I offer the following suggestions:

  1. Abstract and Introduction
    • Clearly articulate the rationale and objectives of the study at the beginning of both the abstract and the introduction.
  2. Syndromic HF-BCC (e.g., Gorlin Syndrome)
    • Include a brief overview of Nevoid Basal Cell Carcinoma Syndrome (NBCCS/Gorlin Syndrome), covering its genetic basis (e.g., PTCH1 mutations) and key clinical features.
  3. Non-Syndromic HF-BCC
    • Add a concise explanation of non-syndromic HF-BCC, including its association with environmental factors such as UV exposure and potential genetic predispositions.
  4. Definition of HF-BCC
    • Clearly define HF-BCC and mention different thresholds used in the literature (e.g., ≥6 BCCs in 10 years).
  5. Broader Context
    • Discuss additional syndromic (e.g., Bazex-Dupré-Christol syndrome, Rombo syndrome) and non-syndromic conditions linked to HF-BCC.
  6. Molecular Mechanisms
    • Expand on the molecular pathways implicated in HF-BCC development (e.g., Hedgehog signaling, DNA repair defects).
    • Include discussion on molecular mechanisms possibly shared with other malignancies (e.g., lymphoma, melanoma, SCC, leukemia, colorectal cancer).
  7. Tumor Distribution
    • Discuss potential reasons why HF-BCCs are less frequently located on the nose and ears, and more often found on the trunk and extremities, compared to low-frequency BCCs.
  8. Histological Patterns
    • Provide insights into why superficial BCC is the predominant subtype in HF-BCC cases.
  9. Results Section
    • Reorganize the results using subheadings to clearly differentiate clinical, histological, and anatomical findings between HF-BCC and 1–2 BCC cases.
  10. Discussion Section
    • Avoid repeating the results; instead, focus on critically interpreting and contextualizing the findings.
    • Discuss the possible biological or environmental mechanisms behind observed trends.
  11. Study Limitations and Future Directions
    • Include a clear acknowledgment of the study’s limitations.
    • Suggest directions for future research to address unanswered questions or further explore the observed associations.
  12. Mechanistic Interpretation of Findings
    • Provide a deeper discussion of the potential mechanisms behind your observed findings—such as the association of HF-BCC with superficial histology, specific anatomical distributions, and comorbid SCC/AK.

Author Response

    1. Abstract and Introduction
      • Clearly articulate the rationale and objectives of the study at the beginning of both the abstract and the introduction.
        Thank you for your remark, we articulated the rationale and the objectives in the abstract and the introduction. 
    2. Syndromic HF-BCC (e.g., Gorlin Syndrome)
      • Include a brief overview of Nevoid Basal Cell Carcinoma Syndrome (NBCCS/Gorlin Syndrome), covering its genetic basis (e.g., PTCH1 mutations) and key clinical features.
        We added information like the genetic basis of this syndrom.
    3. Non-Syndromic HF-BCC
      • Add a concise explanation of non-syndromic HF-BCC, including its association with environmental factors such as UV exposure and potential genetic predispositions.
        Unfortunately, associations with environmental factors such as UV exposure and potential genetic predispositions have not yet been well established in HF-BCC patients, which was one of the motivations for conducting this study. Investigating underlying genetic predispositions will be the focus of future research.
    4. Definition of HF-BCC
      • Clearly define HF-BCC and mention different thresholds used in the literature (e.g., ≥6 BCCs in 10 years).
        The most commonly used definition of HF-BCC in recent literature is the occurrence of 9 or more BCCs within 3 years. Some earlier studies, such as a 2018 study from Stanford, have defined high tumour burden differently (e.g., ≥6 BCCs in 10 years), but these definitions are not used in recent publications.
    5. Broader Context
      • Discuss additional syndromic (e.g., Bazex-Dupré-Christol syndrome, Rombo syndrome) and non-syndromic conditions linked to HF-BCC.
        We added a section comparing the characteristics of HF-BCC patients and other syndromic patients.
    6. Molecular Mechanisms
      • Expand on the molecular pathways implicated in HF-BCC development (e.g., Hedgehog signaling, DNA repair defects).
        There are up to date no molecular pathways known to be implicated in non-syndromic HF-BCC patients, but this is an excellent remark and it will be part of a future study.
      • Include discussion on molecular mechanisms possibly shared with other malignancies (e.g., lymphoma, melanoma, SCC, leukemia, colorectal cancer).
        To date, no molecular pathways have been described for non-syndromic HF-BCC patients. However, once such data become available, comparisons with molecular mechanisms in other malignancies (e.g., lymphoma, melanoma, SCC, leukemia, colorectal cancer) will be an important focus of future research.
    7. Tumor Distribution
      • Discuss potential reasons why HF-BCCs are less frequently located on the nose and ears, and more often found on the trunk and extremities, compared to low-frequency BCCs.
        We adress this in the lines 219-271.
    8. Histological Patterns
      • Provide insights into why superficial BCC is the predominant subtype in HF-BCC cases.
        This is unfortunately not known. Our study is the first one to describe that the superficial BCC in the predominant subtype, but further research is needed to elucidate the pathogenesis.
    9. Results Section
      • Reorganize the results using subheadings to clearly differentiate clinical, histological, and anatomical findings between HF-BCC and 1–2 BCC cases.
        We added tables to address this point.
    10. Discussion Section
      • Avoid repeating the results; instead, focus on critically interpreting and contextualizing the findings.
        Please find attached the improved text.
      • Discuss the possible biological or environmental mechanisms behind observed trends.
        We adress this in the lines 219-271.
    11. Study Limitations and Future Directions
      • Include a clear acknowledgment of the study’s limitations.
        We added a section for this: The markedly higher proportion of HF-BCC patients observed in our dermatological outpatient cohort compared with the prevalence reported in population-based studies suggests a selection bias inherent to specialized clinical settings. While the findings of this study should be interpreted with caution due to the limited sample size of HF-BCC patients (n = 16), they nonetheless provide important preliminary insights into this rare clinical phenotype in a Belgian cohort.  
      • Suggest directions for future research to address unanswered questions or further explore the observed associations.
        Further research will mainly include genetic analysis of non-syndromic HF-BCC patients: The increased occurrence of HF-BCC in patients with actinic keratoses and a history of SCC may reflect a cumulative effect of UV exposure and shared pathomechanisms underlying the development of these skin lesions. Recently, genome-wide association (GWA) meta-analyses have identified over 120 BCC-associated loci, enhancing our understanding of the genetic basis of BCC susceptibility. Moreover, a high degree of association of these BCC-associated loci with SCC risk was observed. Some of these loci were related to pigmentation like SLC45A2, TYR, OCA2 or MC1R. This genetic overlap could explain the association between HF-BCC and personal history of SCC in our study cohort19. Further genetic research in HF-BCC patients is needed to identify associated loci. 
    12. Mechanistic Interpretation of Findings
      • Provide a deeper discussion of the potential mechanisms behind your observed findings—such as the association of HF-BCC with superficial histology, specific anatomical distributions, and comorbid SCC/AK.

        Thank you for your remarks, please find attached the new text, we are open for further remarks and grateful for your help!

Round 2

Reviewer 5 Report

Comments and Suggestions for Authors

None

Author Response

Thank you a lot for your positive feedback!